# Electrical Conductivities of Narrow-Bandgap Polymers with Two Types of π-Conjugated Post-Crosslinking

**DOI:** 10.3390/polym14122472

**Published:** 2022-06-17

**Authors:** Hao-xuan Guo, Hiroshi Takahara, Yusuke Imai, Hiroyuki Aota

**Affiliations:** Department of Chemistry and Materials Engineering, Kansai University, Suita 564-8680, Osaka, Japan; takaharahiroshi895@gmail.com (H.T.); imaiyusuke.425752@gmail.com (Y.I.)

**Keywords:** π-conjugated polymer, narrow-bandgap polymer, post-crosslinking

## Abstract

Bandgap energy is one of the most important properties for developing electronic devices because of its influence on the electrical conductivity of substances. Many methods have been developed to control bandgap, one of which is the realization of conducting polymers using narrow-bandgap polymers; however, the preparation of these polymers is complex. In this study, water-soluble, narrow-bandgap polymers with reactive groups were prepared by the addition–condensation reaction of pyrrole (Pyr), benzaldehyde-2-sulfonic acid sodium salt (BS), and aldehyde-containing reactive groups (aldehyde and pyridine) for post-crosslinking. Two types of reactions, aldehyde with *p*-phenylenediamine and pyridine with 1,2-dibromoethylene, were carried out for the π-conjugated post-crosslinking between polymers. The polymers were characterized by proton nuclear magnetic resonance (^1^H-NMR), thermogravimetric/differential thermal analysis (TG/DTA), UltraViolet-Visible-Near InfraRed spectroscopy (UV-Vis-NIR), and other analyses. The bandgaps of the polymers, calculated from their absorption, were less than 0.5 eV. Post-crosslinking prevents resolubility and develops electron-conducting routes between the polymer chains for π-conjugated systems. Moreover, the post-crosslinked polymers maintain their narrow bandgaps. The electrical conductivities of the as-prepared polymers were two orders of magnitude higher than those before the crosslinking.

## 1. Introduction

π-Conjugated polymers have received a lot of attention because of their versatility in optical and electronic devices such as organic electroluminescent diodes [1,2,3,4], solar cells [5,6,7,8,9,10,11], sensors [12,13,14,15,16], and solid electrolyte capacitors [17,18,19,20,21]. Conventional π-conjugated polymers having various donors and acceptors groups, can generate intramolecular charge transfer by the photoexcitation, can control their absorption to match the solar spectrum, and can be used in the solid-state to achieve high charge carrier mobilities [6,7,8,9,10,11]. In contrast, π-conjugated polymers with doping generate charge carriers (holes and electrons) in the polymer chain and become conductive polymers [22]. The process of doping can introduce a new level between the valence and conduction bands to decrease the bandgap energy (Eg), which allows for an increased mobility of charge carriers between the different energy levels. Sufficiently narrow bandgaps can give rise to high electrical conductivities even in pristine materials. In other words, Eg is one of the most important properties for developing electronic devices because of its influence on the electrical conductivity of substances.

Many methods have been developed to control bandgap [23,24,25,26], one of which is the realization of conducting polymers using narrow bandgap polymers [27,28,29,30,31,32,33,34,35,36,37,38,39,40]. However, the preparation of narrow-bandgap polymers (Eg < 1.0 eV) [41,42] is complex. Previously, we synthesized water-soluble, narrow-bandgap polymers with Eg < 0.19 eV, tuneable in the range 0.3–1.1 eV in aqueous solutions [43,44]. However, their electrical conductivities were low because of the negligible amount of charge carriers in undoped polymers and the inability of charge carriers conducting between the polymer chains. Recently, we used a post-crosslinking reaction (ester crosslinking) to reduce the distance between the polymer chains, which led to increased electrical conductivity [45]. To further increase their electrical conductivity, we focused on π-conjugated post-crosslinking between the polymer chains, which can provide electron conducting routes between them [46,47,48]. On the other hand, solubility control is an important parameter in the preparation of electronic devices using polymers. The stability of soluble polymers thin films in some electronic devices that use electrolyte solutions is poor owing to their resolubility. Post-crosslinking can address this issue by preventing resolubility of the polymer thin films [49,50,51]. Herein, we report the synthesis of water-soluble narrow-bandgap polymers with reactive groups for post-crosslinking by the addition-condensation of pyrrole, sodium salt of benzaldehyde-2-sulfonic acid, and aldehydes containing reactive groups (aldehyde and pyridine). The post-crosslinking step increased the electrical conductivities of the as-prepared polymers by two orders of magnitude, and their resolubility was prevented.

## 2. Experimental Section

### 2.1. Materials

Benzaldehyde-2-sulfonic acid sodium salt (BS), terephthalaldehyde mono (diethyl acetal) (TMDA), *p*-phenylenediamine, and 1,2-dibromoethylene were purchased from Tokyo Chemical Industry (Tokyo, Japan). Pyrrole (Pyr), 4-pyridinecarboxaldehyde (pyridineA) and *p*-toluenesulfonic acid monohydrate (*p*-TS) were purchased from FUJIFILM Wako Pure Chemical Corporation (Osaka, Japan). The Pyr monomer was purified via distillation prior to its use.

### 2.2. Measurements

The UV-Vis-NIR spectra were measured on a JASCO V-670 spectrophotometer, and proton (400 MHz) NMR spectra were recorded on a JEOL ECZ-400S spectrometer. The thermogravimetric/differential thermal analysis (TG/DTA) spectra were measured using a differential thermogravimetric analyzer (Thermo plus TG-DTA 8120, Rigaku, Tokyo, Japan). The sample was scanned from 25 °C to 400 °C with a heating rate of 10 °C/min under air.

For the reduced viscosity (η_sp_/C) measurement, samples were dissolved in phosphate buffer (4.0 g/L sample solution) and measurements were carried out on a viscometer (Ubbelohde-type) at 30 °C. The molecular weights (Mw) of the polymers were calculated using the Mark–Houwink–Sakurada Equation (1):η_sp_/C = kMw^α^(1)
where the constants k (1.16 × 10^−5^) and α (0.894) were approximated from the η_sp_/C and Mw values determined via ultracentrifugal analysis [43]. 

The Eg was calculated using Equation (2):(αhv)^2^ = (const) (hv − Eg)(2)

Here, α is the absorption coefficient. 

The electrical resistances of pressed samples prepared by compressing polymer powder (0.02 g) with a hydraulic press were measured by a four-probe method with a Lorester (MCP-T410, Mitsubishi, Tokyo, Japan). The electrical conductivity of the samples was calculated using Equations (3) and (4): ρ_V_ = R × RCF × t(3)
σ = 1/ρ_V_(4)
where ρ_V_ is the volume resistivity (Ω cm), R is resistance (Ω), RCF is resistivity correction factor, t is thickness (cm), and σ is conductivity (S/cm) [45].

### 2.3. Polymer Synthesis

#### 2.3.1. π-Conjugated Polymers Having Reactive Groups

The non-conjugated polymer **P2** having an acetal group was prepared by reacting a solution of *p*-TS (0.032 g, 0.17 mmol) in DMF (1.0 mL) with another solution of Pyr (0.335 g, 5.0 mmol) and BS (1.041 g, 5.0 mmol) in DMF (5.0 mL) at 10 °C. The mixed solution was stored in the dark. After 24 h, a solution of TMDA (0.104 g, 0.50 mmol) and Pyr (0.034 g, 0.50 mmol) in DMF (2.0 mL) was added to the reaction mixture for 6 h. An aqueous solution of sodium carbonate (5 wt%, 3.2 mL) was added to stop the reaction. Isopropyl alcohol (80 mL) was then added to the reaction mixture. The resulting precipitate, **P2**, was purified by serial reprecipitations, twice from DMF/isopropyl alcohol (8 mL/80 mL), and twice from water/isopropyl alcohol (6 mL/80 mL). The **P2** was finally extracted via freeze-drying (1.296 g).

The π-conjugated polymer, **P4** (with the aldehyde group) was prepared by mixing a solution of DDQ (0.353 g, 1.5 mmol) dissolved in DMF: water (9:1, 6 mL) with a solution of **P2** (0.50 g) in DMF (2.5 mL) at 30 °C. After 6 h, toluene (80 mL) was added to the reaction mixture. Here, the acetal group in the polymer converts to an aldehyde group. The resulting precipitate was purified by a series of reprecipitations from DMF/toluene (8 mL/80 mL) four times, and subsequently from DMF/isopropyl alcohol (8 mL/80 mL), and water/isopropyl alcohol (6 mL/80 mL). The polymer **P4** was finally extracted via freeze-drying (0.461 g).

The polymer **P3** having a pyridyl group (1.011 g) was prepared in a manner similar to that for preparation of **P2**. For the preparation of the π-conjugated polymer, **P5** (having a pyridyl group) was prepared by reacting a solution of chloranil (0.670 g, 2.7 mmol) dissolved in DMF (14 mL), with another solution of **P3** (0.70 g) dissolved in DMF (4.0 mL) at 30 °C. After 6 h, toluene (80 mL) was added to the reaction mixture. The resulting precipitate was purified by serial reprecipitations similar to **P4.** The polymer **P5** was finally extracted via freeze-drying (0.663 g).

#### 2.3.2. Two Types of π-Conjugated Post-Crosslinking Reactions

*Type A: Imine type*.

**P6** was prepared as follows: *p*-TS (0.20 g, 1.0 mmol) was added to a solution of **P4** (with the aldehyde group; 0.20 g) and *p*-phenylenediamine (0.20 g, 1.8 mmol) in DMF (3 mL). The resulting mixture was stirred at 100 °C. After 24 h, the insoluble fraction was separated by suction filtration and washed with water and methanol. **P6** was obtained via vacuum drying (0.088 g). 

*Type B: Quaternized pyridine type*.

**P7** was prepared as follows: 1,2-dibromoethylene (0.71 g, 3.8 mmol) was added to a solution of **P5** (having a pyridyl group; 0.30 g) in DMF (3 mL). The resulting mixture was stirred at 100 °C. After 48 h, the insoluble fraction was separated by suction filtration and washed with water and methanol. **P7** was obtained by vacuum drying (0.104 g).

## 3. Results and Discussion

### 3.1. Polymerization

The preparation of the polymers is shown in Figure 1. The addition–condensation polymerization of Pyr with aldehyde monomers BS, TMDA, or pyridine was performed in DMF at 10 °C to obtain the non-conjugated polymers (**P2**, **P3**). **P2** and **P3** were then oxidized to π-conjugated polymers (**P4**, **P5**). Incidentally, the TMDA in **P2** reacted with the oxidant to give an aldehyde via acetal hydrolysis (TMDA). **P2**–**P5** were soluble in water, methanol, and DMF, and insoluble in isopropyl alcohol, acetonitrile, THF, acetone, chloroform, and hexane. The reduced viscosities and molecular weights of the polymers are summarized in Table 1. The reduced viscosities of **P4** and **P5** are greater than those of **P2** and **P3** because π-conjugation makes the polymer chain rigid. Figure 1 shows the ^1^H-NMR spectra of the polymers (**P2**–**P5**). For **P2** and **P3** (the nonconjugated polymers), the peaks at 9–10 and 5.5 ppm are assigned to the Pyr protons (a and b), the peak at 6.5 ppm is assigned to the proton (c) between Pyr and BS, and the peaks at 7–8 ppm are assigned to the BS protons (d and e). The peaks at 1.3–1.6 ppm are assigned to the protons in the diethyl acetal group (g) of **P2**. The peaks at 8.3–8.5 ppm are assigned to the protons of the pyridine group (g) in **P3**. After the oxidation reaction, peak broadening can be observed from 5.5 to 8.5 ppm. This can be attributed to the conversion to a π-conjugated system, which causes the polymer chain rigid owing to the slow molecular motion. In the case of **P4**, due to acetal (TMDA) hydrolysis by the oxidant, the peaks at 9.6–10.4 ppm (h) corresponding to aldehyde protons appear. However, for **P5**, the pyridine protons (f) are not assigned because of overlapping peaks and broadening peaks. Due to the low number of reactive groups introduced, the TMDA, pyridine, and aldehyde proton signals are relatively weak. To prove that the reactive groups can be introduced into the polymers, a polymer was synthesized by the same synthetic method using 1-pyrenecarboxaldehyde, which can be spectroscopically monitored. The absorption spectrum of this material confirms the introduction of pyrene units into the polymer (Appendix A). Polymers with a large number of reactive groups were also synthesized as described in the Appendix A. Appendix A show the ^1^H-NMR spectra of polymers with a large number of reactive groups. The TMDA, pyridine, and aldehyde protons are detected at the same ppm (Figure 1).

A post-crosslinking reaction was performed as described in Section 2.3. The post-crosslinked polymer **P6** was prepared by imine-crosslinking reaction of **P4** and *p*-phenylenediamine. The post-crosslinked polymer **P7** was prepared by quaternization of pyridine crosslinking reaction of **P5** and 1,2-dibromoethylene. **P6** and **P7** were insoluble in almost all the solvents (water, methanol, ethanol, isopropyl alcohol, acetonitrile, DMF, THF, acetone, chloroform, and hexane). Resolubility of the polymers was prevented by post-crosslinking.

### 3.2. UV−Vis-NIR Absorption Spectroscopy

Figure 2 and Appendix A show the UV-Vis-NIR spectra of **P2**–**P5** dissolved in phosphate buffer (pH 6.9) to prevent self-doping. The longer-wavelength absorptions are due to the π–π* excitation of the expanded π-conjugation. The bandgaps of the polymers calculated from their absorption are listed in Table 1. The synthesized conjugated polymers were narrow-bandgap polymers (Eg < 1 eV) [42].

As the post-crosslinked polymers were insoluble in almost all the solvents, the absorption spectra of the post-crosslinked polymers dissolved in solvent could not be obtained. The diffuse reflection of the polymer powders was measured instead (Appendix A), and the UV-Vis-NIR spectra (Figure 3) of the powdered polymers were obtained using the corresponding diffuse reflection spectra. Compared to the non-conjugated polymers (**P2** and **P3**), the post-crosslinked polymers (**P6** and **P7**) were similar to the π-conjugated polymers (**P4** and **P5**) having a broad absorption band. After the post-crosslinking reaction, **P6** and **P7** maintained their narrow bandgaps.

### 3.3. Thermal Stability and Electrical Conductivity Measurements

The thermal stability of the post-crosslinked polymers was investigated via TG/DTA under air. The post-crosslinked polymers **P6** and **P7** recorded an onset decomposition temperature of 300 °C and exhibited a high heat resistance (Appendix A).

A comparison of all the pressed polymer samples (Table 2) reveals that the post-crosslinked polymers show a higher electrical conductivity than those without crosslinking. The conductivity of **P6** and **P7** is 124 and 203 times higher than that of **P4** and **P5**, and significantly higher than the conductivity of our previously studied ester-crosslinked polymers [45]. The improved electrical conductivity occurs due to an improvement in the electron-conducting routes (π-conjugated) between the polymer chains via post-crosslinking (imine-crosslinking and quaternization of pyridine-crosslinking). In addition, **P7** has the highest conductivity, because its charge carriers were generated during the crosslinking reaction between the polymer chains (quaternization of pyridine).

## 4. Conclusions

In conclusion, we synthesized novel, reactive, narrow-bandgap polymers with reactive groups, aldehyde and pyridine, via the addition–condensation of Pyr, BS, and TMDA/pyridineA. Imine-crosslinking via the aldehyde group with *p*-phenylenediamine and quaternization of pyridine with 1,2-dibromoethylene were used for the post-crosslinking reaction between the polymer chains. The post-crosslinked polymer was insoluble, maintained a narrow bandgap, and showed a high heat resistance. Post-crosslinking develops π-conjugated, electron-conducting routes between the polymer chains, and charge carriers are generated in the polymer via the quaternization of pyridine. The electrical conductivities of the as-prepared polymers were two orders of magnitude higher than those before the crosslinking. These polymers have great potential in the field of materials science, particularly for the development of novel solid electrolyte capacitors.

## Data Availability

Not applicable.

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
