# Peer review of "Electrical Conductivities of Narrow-Bandgap Polymers with Two Types of π-Conjugated Post-Crosslinking"

_polymers, 2022, doi:10.3390/polym14122472_

Round 1

Reviewer 1 Report

In this work, water-soluble narrow bandgap polymers have been synthesized, to which post-crosslinking has been applied, which has promoted the formation of side chains, a strategy that has obtained two technical advantages: the electrical conductivity of the the polymer films obtained, by creating shortest paths for electric current at the molecular level; and at the same time, the resolubilization of the thin conductive films in the electrolytic solutions with which they are in contact in certain electronic devices has been prevented, increasing the durability of the electronic component that is manufactured with these polymers.

As the manufacture of electrical signal measurement and transmission devices with organic-based active elements is one of the great strategic objectives at a global level in recent years, combined with the detailed structural exposure at the molecular level to explain the basis of the improvement in the electrical response of the prepared polymers that is carried out in the work, in my opinion the scientific and technological interest of this article is beyond any doubt.

The methodology for the synthesis and subsequent crosslinking of the prepared polymers is clearly and in detail described, and also correctly referenced. Also the procedural technique of the tests to determine the molecular weights of the different polymers prepared is correctly described. And the instrumental techniques used to carry out the structural analyzes at the molecular level of the polymers are of a very high level, and all the operative methodologies and the corresponding discussions of the results are of a good scientific level, and are presented clearly and following an adequate thread. informative driver for a good understanding by the reader.

The formal presentation of the work seems correct to me, and follows a logical structure. The tables are well prepared, as they clearly show the most important parameters and effects that they are intended to highlight. The images are clear, and the size of the numbers and letters of the axes, and the legends of the graphs seem to me sufficient for an adequate visualization, with one exception: the letters that appear inside the graph corresponding to Fig. 1 (dealing with NMR spectra) are too small for comfortable viewing, they should be enlarged a bit.

I have not detected any appreciable formal error, except for some words that are very badly "cut" from the orthographic point of view, by a hyphen at the end of the line (for example, at the end of line 235, "pol-" appears, and at the beginning of line 236, Aparce " ymers ", and it looks very bad). But I understand that this effect will be solved in the layout of the article

As regards the quality of the English wording of the manuscript, I do not consider myself qualified to assess it. I always recommend a thorough spelling and syntax check. Although the background and general meaning of the wording suggests a sufficiently interesting presentation for the reader versed in the field of work. The approach and definitions that comprise the introduction of the work have a correct order of presentation and are adequately referenced. And as for the conclusions of the work, they seem correct to me and are consistent with the objectives declared in the introduction.

Now, at the end of the conclusions, the following sentence appears: "These polymers have great potential in the field of materials science, in particular for the development of new solid electrolyte capacitors." So, I must understand that the prepared polymers, provide the first phase of development of the active element of an organic-based condenser. It seems to me an excellent proposal, but in this case, in my opinion, the work requires an electrical response test of the prepared polymers that has not been presented: a cyclic voltammetry test of the polymer film in an electrolytic cell, which verifies its capacity charge storage redox. In addition, this test will be suitable for testing the insolubility of polymers in a typical electrolyte (saline solution) in a usual range of working potentials for this type of material (for example, if it is an aqueous medium, between –0.5V and +1.0V).

So, in conclusion, in addition to the small change in font size that I have indicated for the graph in Fig. 1, I urge the authors to perform the cyclic voltammetry test with at least one of the two post-crosslinked polymers that have prepared, as a precondition for this interesting work to be suitable for publication in this journal.

Author Response

Dear Reviewer:

Reviewer 2 Report

The authors present new polymers, which have great potential in the field of materials science, particularly for the development of novel solid electrolyte capacitors and solar cells.  The paper could be accepted after revision. -Molecular weight of linear polymers should be presented. -Is it possible to measure glass transition temperature of the polymers?  The DSC results should be presented. - Application of the polymers in solid electrolyte capacitors or in solar cells should be demonstrated.

Author Response

Dear Reviewer:

Round 2

Reviewer 1 Report

The authors have implemented my formal recommendations in the manuscript, and have also provided a reflection on how the prepared polymers may be good candidates for the future fabrication of organic-based electrolytic capacitors, reinforcing their statement in the conclusions.

Despite the fact that no cyclic voltammetry test could be provided, I understand that in future works this electrical response characterization test will be added. I consider that the discussion presented by the authors has sufficient argumentative quality to accept their manuscript. for publication in this journal.

Reviewer 2 Report

If editor and other reviewers agree I could also recommend the paper for publication after the revision.